# Position: Fodor and Pylyshyn's Legacy —
# Still No Human-like Systematic Compositionality in Neural Networks

## Abstract

The strength of human language and thought lies in their ability for systematic compositionality: the meaning of a unit (semantics) can be inferred from its structure (syntax). While Fodor and Pylyshyn famously posited that neural networks inherently lack this capacity and in turn are no viable model of the human mind, Lake and Baroni recently presented meta-learning as a pathway to compositionality. In this position paper, we critically evaluate this claim, highlighting limitations in the proposed framework of meta-learning for compositionality (MLC). Specifically, we identify a class of test cases compatible with Lake and Baroni's setup that consistently provoke transduction errors despite falling well within the scope of human-like abilities. We further identify overlooked yet essential elements required for substantive claims of systematic generalization. Therefore, despite the success of neural models in mimicking human behavior, it seems premature to claim that modern architectures have overcome the limitations raised by Fodor and Pylyshyn. This issue is pivotal to the AGI debate, as systematic generalization is crucial for human-like reasoning and adaptability.

## 1. Introduction

Meta-learning, or *learning to learn* from different situations, is an interesting challenge closely related to human intelligence. It is a core element of our educational system that we learn *how to learn* without explicit prior knowledge about each situation in life, as their variations are manifold. Similarly, the use of language embodies this adaptability, requiring the integration of learned rules with contextual nuances

[1]Anonymous Institution, Anonymous City, Anonymous Region, Anonymous Country. Correspondence to: Anonymous Author <anon.email@domain.com>.

Preliminary work. Under review by the International Conference on Machine Learning (ICML). Do not distribute.

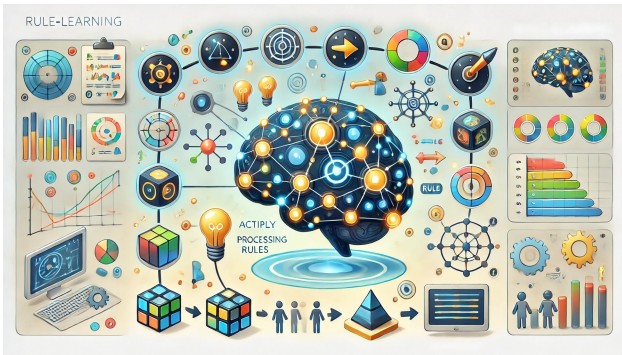

*Figure 1.* "A conceptual illustration visualizing rule-learning, without any text" by DALL·E; some semantics of *text* seem to be misunderstood.

to navigate both familiar and novel scenarios. Language exemplifies how humans apply systematic generalization, seamlessly combining learned grammatical structures and vocabulary to create and interpret new expressions. This dynamic interplay between rules and context bridges the abstract principles of meta-learning with the practical mechanisms that underlie communication and cognitive reasoning.

The principle of compositionality is a key challenge for artificial neural networks, as it requires the ability to develop systematic representations and behavior. Unlike humans, neural models often struggle to generalize such rules (Nezhurina et al. 2024, Wüst et al. 2024a, Bayat et al. 2025) across contexts, reflecting fundamental gaps in their representational and operational frameworks. As artificial neural networks are constrained by their reliance on finite representational spaces and distributed encoding schemes, these limitations manifest in their difficulty in consistently applying composition rules across different scenarios. While humans can effortlessly recombine learned concepts to interpret novel sentences or solve unique problems, neural networks lack the inherent transparency, flexibility, and reflexivity to perform similar feats. Their opacity, driven by distributed representations, hinders their ability to systematically manipulate components and infer relationships.

Lake & Baroni (2023) introduced a meta-learning frame-

work attempting to mitigate these challenges by introducing episodic training tasks that require rule inference. The framework involves presenting neural networks with support examples governed by hidden grammars and testing their ability to generalize these rules. This episodic approach aims to train networks for systematic generalization, using meta-learning principles to approximate human-like reasoning. They claimed to overcome some fundamental limitations of neural networks, prominently stated by Fodor & Pylyshyn (1988). However, there is also plenty of evidence of the limitations of modern deep learning models with human-like capabilities in language understanding that rely on systematic compositional reasoning (Deletang et al. 2023, Zhang et al. 2023, Dziri et al. 2024, Mészáros et al. 2024, Bayat et al. 2025), and we provide further insights that even Lake and Baroni's model fails to prove its systematic behavior in several instances.

Despite its potential, the framework's reliance on learned distributions and predefined rule forms limits its scope. Generalization remains limited to permutations of known rules rather than discovering entirely new principles. The difficulty of scaling to complex tasks with deeper nesting further underscores the persistent gaps in achieving true human compositional reasoning. Lake and Baroni's framework provides valuable insights, but also highlights the need for innovation in training and evaluating neural networks to overcome these limitations, since behavioral similarities may mask fundamental differences in underlying mechanisms.

Thus, we argue in this paper that: **Neural networks have not yet achieved learning systematic compositional skills.**

We derive this position as follows:

- We locate the nature of Lake and Baroni's approach in refuting Fodor and Pylyshyn's claims that neural networks cannot reliably develop *compositional representations* and *structure-sensitive operations*.

- We show that within their setup, the model exhibits various non-systematic behaviors that can not be considered human-like.

- We argue for several necessary aspects of training and evaluation of meta-learning systems to achieve and assess their systematicity.

- We adapt Fodor and Pylyshyn's arguments in light of the modern development of deep-learning systems to argue for a future of models capable of learning symbolic representations for artificial cognition and representation learning.

## 2. The Challange of Compositionality

### 2.1. Fodor and Pylyshyn's Legacy

In their influential 1988 paper, *Connectionism and cognitive architecture: A critical analysis*[1], Fodor and Pylyshyn claim that artificial neural models are unsuitable for modeling the human mind on a cognitive level. They review several arguments for the combinatorial structure of mental representations, highlighting the systematicity of these representations that follow the compositional nature of cognitive capabilities; the ability to understand some given thoughts implies the ability to understand various thoughts not only with semantically related content but also of more combinatorial complex structure. Nevertheless, they also consider the possibility that artificial neural networks may play a role in *implementing* cognition.

**Differentiating neural networks and symbolic systems.** Their work starts with discussing the disagreement about the nature of mental processes and mental representations between the so-called Connectionist approach, which focuses on artificial neural networks, and the Classical approach, favoring symbolic systems like Turing Machines for modeling cognitive capacities. They stress that it is neither about the explicitness of rules, nor the reality of representational states, nor about nonrepresentational architecture, as a "Connectionist neural network can perfectly well implement a Classical architecture at the cognitive level."[2] While both "assign semantic content to *something*"[3], it is identified as the central difference that they disagree about what primitive relations hold among these content-bearing entities. The sole importance of causal connectedness in neural networks is contrasted with a range of semantic and structural relations in symbolic systems. Only the sensitivity for both semantic and structural relations is expected to allow for commitment to the compositionality of mental representations with combinatorial syntax and semantics. Furthermore, the operations the models perform in transforming one representation into another are sensitive to the structure of these representations and not only their semantics.

**Productivity, compositionality and systematicity of cognitive ability.** The need for these two properties of symbol systems, *compositional representations* and *structure-sensitive operations*, is justified by "three closely related features of cognition: its productivity, its compositionality, and its inferential coherence."[4] Only structure-sensitive operations in combination with a combinatorial structure and semantics of representations can explain the (under appropriate idealization) *unbounded capacities* of a representational

---

[1](Fodor & Pylyshyn, 1988).
[2]Ibid., p.11.
[3]Ibid., p.12, emphasis in original.
[4]Ibid., p.33.

system. Similarly, cognitive capacities are systematic in that the capability for producing or processing some representations is syntactically connected to the capability for producing or processing certain other representations without relying on processing every specific semantics, e.g. understanding the form of the expression $A \wedge B \Rightarrow A$ implies the capability to understand the expression for any substituents of $A$ or $B$. In fact, systematicity makes a stronger by using a weaker assumption as, "[p]roductivity arguments infer the internal structure of mental representations from the presumed fact that nobody has a *finite* intellectual competence [and by] contrast, systematicity arguments infer the internal structure of mental representations from the patent fact that nobody has a *punctuate* competence." [5] Closely related to systematicity is the compositionality of mental representations since capabilities for producing or processing representations can be linked not only syntactically but also semantically. Here, it is important to note that not every mental representation is expected to be compositional, e.g., the understanding of some expressions in natural language, as the "similarity of constituent structure accounts for the semantic relatedness between systematically related sentences only to the extent that the semantic properties of the shared constituents are context-independent." [6] A last cognitive feature is the systematicity of inference. Recalling the example of conjunction $A \wedge B$ entailing its constituent $A$, it is not only the mental representation of understanding this rule that is systematic but also its application for coherent inference between thoughts, demanding again for the structure-sensitivity of operations in symbolic systems.

**Neural networks for implementing symbol systems.** Finally, Fodor and Pylyshyn comment on treating Connectionism as an implementation theory for cognitive architecture. They "have no principled objection to this view" [7]. Still, they stress that when neural networks are only a *method for implementing* cognitive architecture, their internal states are useless for understanding the nature of mental representations and, therefore, "irrelevant for the psychological theory" [8]; neural networks would be just of neurological means, and the need for and relevance of symbol systems for modeling cognition would stay untouched.

### 2.2. Lake and Baroni's Objection

**Compositional seq2sec tasks.** Lake and Baroni present their work as evidence against Fodor and Pylyshyn's claims. They introduce a meta-learning framework that they claim achieves or exceeds human-level systematic generalization across their evaluations. Their experimental setup is based

---

[5]Ibid., p.40, emphasis in original.
[6]Ibid., p.42.
[7]Ibid., p.67.
[8]Ibid., p.65.

on sequence-to-sequence (sec2sec) transduction tasks, They consider sequences generated over 8 pseudolanguage tokens $u \in U$ for the input domain $X = U^*$, while the output domain $Y = C^*$ comprises sequences generated over 6 different color tokens $c \in C$. Both domains are linked by a transduction grammar, i.e., a set of production rules that define how a sequence of input tokens is translated into a color sequence. Each rule is of two sorts; it can state a primitive transduction rule $u \to c$, simply mapping an input token to an output token; otherwise, it states a unary operation $v_1 u \to f_u(v_1)$ or binary operation $v_1 u v_2 \to g_u(v_1, v_2)$ where any $f$ is some $n$-fold ($n \leq 8$) repetition, any $g$ is some combination of repetition, permutation and concatenation. Each $v_i$ is either a single token $u_i$ or the whole proceeding or succeeding token string $x_i$. By the iterative composition of these rules, such a grammar generates a set of translatable input sequences $\bar{X} \subseteq X$.

**Seq2seq meta-learning framework for evaluation of human systematic generalization.** With these transduction tasks, Lake and Baroni set up a meta-learning framework with *EPISODES* associated with different transduction grammars. Each episode combines a *SUPPORT* set of input-output transduction pairs and a *QUERY* set of input-output pairs, while any pair is consistent with the associated grammar. The query outputs are hidden, and it is the task to replicate them with the support examples as the only information given; the underlying transduction grammar also remains hidden. In this way, explicitly inferring the grammar rule is unnecessary. Still, the capability to implicitly extract or hypothesize the actual grammar rules is expected to be essential for reliably deriving the correct query outputs. A standard seq2seq transformer network is now trained on query examples of various episodes. The transformer encoder processes a query input combined with the support pairs of its episode as context, and the transformer decoder generates an output sequence.

## 3. Systematicity through Meta-Learning

**Locating Lake and Baroni's approach.** In order to evaluate the proposed framework for systematic generalization by meta-learning neural networks with respect to Fodor and Pylyshyn's claims, we will first clarify which of Fodor and Pylyshyn's arguments Lake and Baroni are referring to, since they primarily present an implementation of what they claim is a human-like systematic capability, but directly address a challenge. They themselves situate their work as a contribution to the line of argument that Fodor and Pylyshyn's statements no longer apply to current model architectures; they are not criticizing the properties of human cognition, but the alleged inability of neural networks to reliably develop *compositional representations* and *structure-sensitive operations*. By focusing on behavioral tests rather

than ablation studies that directly examine the structure of learned representations, Lake and Baroni emphasize the *structure sensitivity* and *systematicity* of their model, which is crucial for demonstrating compositional abilities and coherent behavior. Furthermore, they present their meta-learning framework for compositionality to systematically train neural networks with these abilities. While a single neural network with compositional abilities would not contradict Fodor and Pylyshyn, who did not claim any *limits on implementability* of cognitive abilities, a framework that reliably archives *compositional abilities by stochastic learning* methods would actually contradict their main point of criticism. Unfortunately, we will see in the following section that the model trained on meta-learning still fails to reliably demonstrate compositional ability in several examples.

### 3.1. Examining the Lack of Compositionality

Lake and Baroni mention that generalization beyond training only occurs with respect to *new combinations* of three grammar rules out of the same set of grammar rules used during training. However, when we account for invariance under the atomic assignments of colors to language tokens and the mere labeling of operations, we find that 179/200 validation episodes have a combination of non-primitive grammar operations that were already within the 100000 training episodes. So, even if the model would achieve highly systematic results on the test episodes, it could be just due to memorization of the experienced operation patterns and learning to extract the correct labels out of the episode's support examples. However, we can even show that there is non-systematic behavior within their repository of testing episodes; we reevaluate their pre-trained $'net - BIML - top'$ model on the same set of $'algebraic'$ test episodes with the mere difference that we did 10 evaluations of all query examples for every testing episode, for statistical purposes, similar to the one episode they further evaluated against human performance. We find that the model performs worst on episodes #133, #32, and #122, with accuracies of only 41%, 52%, and 54% on the query examples, respectively. (See next paragraph and Appendix 7 for details.)

**Failure in rule extraction.** Further investigation of Episode #133 (see Table 1) reveals that the model struggles to correctly process the semantics of the language token ⟨fep⟩ with the hidden grammar rule $x_1$ fep $\rightarrow x_1\ x_1$ and will therefore refer to as ⟨twice⟩ and mixes it up with the token ⟨gazzer⟩ (with $x_1$ gazzer $\rightarrow x_1\ x_1\ x_1$) we will call ⟨thrice⟩. It seems to have a problem with the sole example featuring ⟨twice⟩, ⟨■ thrice twice → ●●●●●●⟩, which also happens to contain ⟨thrice⟩. But since ⟨thrice⟩ has several iconic examples in the support, it is expected that a reasoner with compositional skills will be able to systematically use a single example and remain consistent

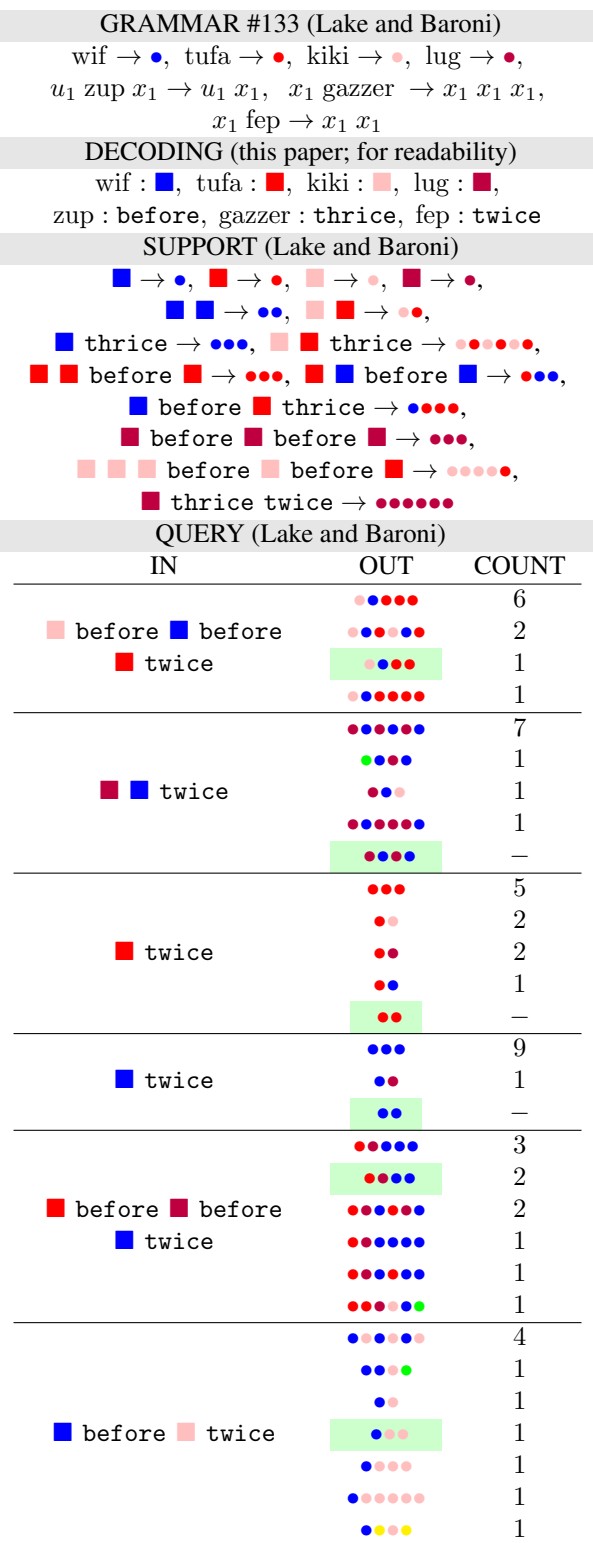

*Table 1.* Episode #133 with 10 evaluations for each query example; SUPPORT and QUERY are decoded for better readability. Expected outputs backed with green. The model shows *incoherent processing* and *systematically mistakes* twice for thrice. Further results can be found in Appendix A.1. (Best viewed in color.)

with the rest of the support information. By considering the examples $\langle\blacksquare \to \bullet\rangle$, $\langle\blacksquare \to \bullet\rangle$, $\langle\blacksquare\ \texttt{thrice} \to \bullet\bullet\bullet\rangle$, human systematicity would at least suspect some semantics of $\langle\texttt{twice}\rangle$ that are different to those of $\langle\texttt{thrice}\rangle$.

**Non-systematic parsing.** Interestingly, the hidden grammar allows for an ambiguous interpretation of nested transduction queries, which would actually be a challenge for a systematic reasoner. For instance, the query $\langle\blacksquare\ \texttt{before}\ \blacksquare\ \texttt{twice}\rangle$ could be parsed as either $\langle\blacksquare\ \texttt{before}\ (\blacksquare\ \texttt{twice})\rangle$ (marked as target by Lake and Baroni ) or $\langle(\blacksquare\ \texttt{before}\ \blacksquare)\ \texttt{twice}\rangle$, and similarly for a query with $\langle\texttt{thrice}\rangle$. But the support example $\langle\blacksquare\ \texttt{before}\ \blacksquare\ \texttt{thrice} \to \bullet\bullet\bullet\bullet\rangle$ should at least induce a bias toward the intended processing. But the responses to this challenge also lack systematicity; while the frequent mistakes $\langle\blacksquare\ \texttt{before}\ \blacksquare\ \texttt{before}\ \blacksquare\ \texttt{twice} \to \bullet\bullet\bullet\bullet\bullet\rangle$ and $\langle\blacksquare\ \texttt{before}\ \blacksquare\ \texttt{before}\ \blacksquare\ \texttt{twice} \to \bullet\bullet\bullet\bullet\bullet\rangle$ could be explained by processing $\langle u_1\ \texttt{before}\ (u_2\ \texttt{before}\ (u_3\ \underline{\texttt{thrice}}))\rangle$ while, in contrast, a similar explanation to the the error $\langle\blacksquare\ \texttt{before}\ \blacksquare\ \texttt{twice} \to \bullet\bullet\bullet\bullet\bullet\rangle$ would be the parsing $\langle(u_1\ \texttt{before}\ u_2)\ \underline{\texttt{thrice}}\rangle$. We will further discuss the importance of systematicity for meta-learning systems in Section 3.2.

**Violating structure-sensitivity.** Besides both previous failure modes that are related to incompetence in extracting information from the support examples, we also found query examples for episode #1 that reveal additional non-systematicity (see Table 2 in Appendix 7 for extended version). For queries with patterns $\langle u_1\ \texttt{thrice}\ \texttt{around}\ u_2\ u_3\rangle$ and $\langle u_1\ \texttt{around}\ u_2\ u_3\ \texttt{twice}\rangle$ we first see that the model never parses $\langle\texttt{around}\rangle$ as intended. Instead of $\langle((u_1\ \texttt{thrice})\ \texttt{around}\ u_2)\ u_3\rangle$ and $(((u_1\ \texttt{around}\ u_2)\ u_3)\ \texttt{twice}\rangle$ the stable outputs can be explain with parsing $\langle\texttt{around}\rangle$ as intended. Instead of $\langle(u_1\ \texttt{thrice})\ \texttt{around}\ (u_2\ u_3)\rangle$ and $(\langle u_1\ \texttt{around}\ (u_2\ u_3))\ \texttt{twice}\rangle$ — except for the cases, $\langle\blacksquare\ \texttt{thrice}\ \texttt{around}\ \blacksquare\ \blacksquare\rangle$, $\langle\blacksquare\ \texttt{thrice}\ \texttt{around}\ \blacksquare\ \blacksquare\rangle$, $\langle\blacksquare\ \texttt{around}\ \blacksquare\ \blacksquare\ \texttt{twice}\rangle$, *where it would not make any difference*! Only the (also ambiguous) case $\langle\blacksquare\ \texttt{around}\ \blacksquare\ \blacksquare\ \texttt{twice}\rangle$ is correctly processed in 6/10 cases — however, with even worse performance than on the unambiguous examples. Despite the structural similarities to the other query examples up to the color combination, we see a non-systematic deviation in responding that leaves compositional skills in doubt.

**Limits in productivity.** Finally, we want to point out that Lake and Baroni's setup only enables the model to process input sequences of up to 10 tokens and generate output sequences of up to 8 color tokens (which further restricts the admissible input sequences). This limits the possibilities for testing more complex input sequences and thus assessing

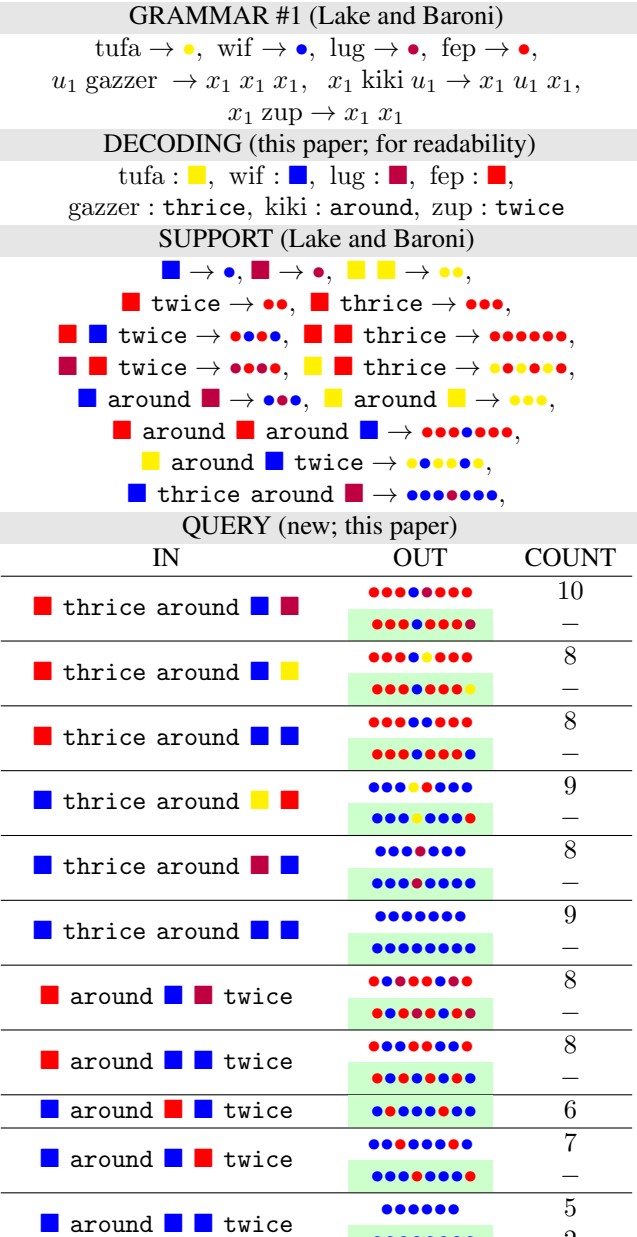

*Table 2.* Episode #1 with our own query examples and with 10 evaluations for each input; SUPPORT and QUERY are decoded for better readability. Expected outputs backed with green. Further results can be found in Appendix A.4 (Best viewed in color.)

the *productivity* for the model's skill.

## 3.2. Our Position on Meta-Learning Systems

We now discuss whether meta-learning, beyond Baroni's framework, could be a promising approach towards human-like compositional skills, despite the demonstrated limitations in the specific setup. Meta-learning systems aim to emulate human-like learning by incorporating systematic-

ity and flexibility into their architectures. These systems aim to (1) generalize beyond training examples by inferring composition rules from limited examples, (2) adapt to novel contexts with flexibility as a key expectation, allowing systems to quickly transferring skills to new domains with minimal retraining, and (3) mirror human-like cognition by ensuring that error patterns and reasoning paths are still systematic, explainable, or even self-correcting.

**Weakness of non-reflective training.** One of the primary shortcomings in Lake and Baroni's work is the employment of a one-shot prediction approach. Models are trained to perform a direct transduction without intermediate reflection or validation steps on the presented support examples. To guarantee systematic production of outputs, we claim that it is of primary importance for meta-learning models to iteratively extract, *validate, and correct* their current belief in the extracted rules. In the previous section we showed that the models of Lake and Baroni fail to validate extracted rules against the support and, therefore, systematically fail to correctly extract (and in consequence apply), e.g., the `twice` rule.

**Focus on systematicity rather than productivity.** Considering the role that underlying grammars play with regard to meta-learning or specifically non-meta-learning problems, any of today's modern transformer systems can be broken by providing them with more and more complex problems, up to a point where the models are no longer expressive enough to comprehend the problem as a whole. This could be, for example, due to the depth of rule nesting or simply due to the length of the input. While the general ability to learn to transcribe rules certainly is a prior requirement of meta-learning systems in the particular discussed setting, one would not necessarily deny such systems the ability to perform meta-learning reasoning even when failing to accomplish such tasks due to the aforementioned reasons. When talking about meta-learning tasks, one is not so much interested in the ability to derive rules of arbitrary complexity —which rather constitutes a problem of classical machine learning— but in the ability of these models to systematically discover, verify, apply, and combine said rules or to systematically learn from its mistakes. When comparing to human reasoning (Nezhurina et al., 2024; Wüst et al., 2024b), meta-reasoning abilities are not judged by the ability to produce transductions in a one-shot fashion, but rather with a focus on the result being correct in the *final output*. We, therefore, formulate the following claim:

*Claim* 1. A characteristic of *successful* meta-learning systems is the ability to consistently abstain from *non-systematic* errors.

Our claim primarily regards the consistency of model behavior and we, therefore, differentiate between systematic and non-systematic errors. Systematic errors might arise

due to wrong inherent assumptions of the model that, then however, get systematically applied in consequence. Such errors might stem from wrong assumptions on the general task setup. In our setting, this might involve assumptions about the unique interpretation of rules –see, e.g., our discussion on possibly ambiguous rule interpretations in Lake and Baroni–, and in general might be due to exogenous factors and implicit assumptions that where not be captured during training phase. While such errors might not produce the desired output they follow a systematicity that give rise to the assumption that the model might have been able to learn the right rules, given the correct underlying assumptions. The absence of systematicity, however, poses a much larger fault. Here, models might expose erratic 'glitches', resulting in a non-human-like behavior, that is absent of any systematicity. As the underlying reasons for such behavior might not be understood in general, it is unclear how to treat and correct such errors.

Last, we derive two positions regarding essential aspects of evaluation and training of successful meta-learning systems:

**Position on Evaluation.** Assessing and postulating systematic or compositional skills in neural networks requires either the direct evaluation of the model's internal representations, which would require an inspectable or explainable network architecture, or the use of comprehensive ablation studies that systematically testing a model's behavior in out-of-distribution situations.

**Position on Implementation and Training.** In order to obtain compositionality and systematicity within the discussed meta-learning tasks, the presence of symbolic representations within neural networks is vital to ensure consistent application and composition of rules. We want to emphasize that while Fodor and Pylyshyn remain unrefuted in the general analysis, today's discussion of modern neural network architectures continuously evolve to develop symbolic representations e.g. in the form of circuits (Olah et al., 2020; Wang et al., 2022; Conmy et al., 2023; Hanna et al., 2024). These explicit representations are important building blocks that promote consistent behavior and allow for explicit reflection and iterative correction of possible inconsistencies of the extracted rule sets. Last, it is important to mention that reflective behavior is likely to not evolve from training on one-shot transduction tasks, but *requires models to have the possibility iterate, validate and correct over the extracted rule sets*. Most recently important breakthroughs in this direction have been achieved in RL training of language reasoning models (Stiennon et al., 2020; Ouyang et al., 2022; Bai et al., 2022; Lee et al., 2023; DeepSeek-AI et al., 2025).

## 4. Related Work

**Human-like compositionality.** Regarding the importance of compositionality for cognitive skills Fodor & Lepore (2001) and Fodor (2001) extend the disscusion of Fodor & Pylyshyn (1988) on the compositional nature of language and thought. While (natural) language incorporates some non-compositional structures due to *context sensitivity*, compositionality is argued to be essential for (a language of) thought. This falls in line with more recent work of Fedorenko et al. (2024) trying to find evidence for how language is primarily a tool for communication rather than thought.

**Compositionality in neural networks.** Besides Lake & Baroni (2023), there is older as well as recent work trying to demonstrate compositional or meta-learning skills achieved with neural network architecture (Botvinick & Plaut, 2004; Santoro et al., 2016; Park et al., 2024; DeepSeek-AI et al., 2025). Other work is proposing frameworks for learning and assessing compositional skills (Petrache & Trivedi, 2024; Sinha et al., 2024) or other intelligent behavior (Chollet, 2019) and Bayat et al. (2025) is introducing memorization-aware training to tackle overfitting to spurious correlation encountered in training.

**Limitations in systematicity.** Several works evaluate and demonstrate the limitations of modern AI models in compositional or systematic generalization tasks (Bender et al., 2021; Deletang et al., 2023; Dziri et al., 2024; Mészáros et al., 2024; Nezhurina et al., 2024; Zhang et al., 2024; Wüst et al., 2024b) and there is also another targeted response to Lake and Baroni's work, presenting problems of non-systematic behovior (Goodale & Mascarenhas, 2023).

**Importance of symbolics.** There is also more recent work that stresses the importance of symbolics. Ellis et al. (2020) introduces a machine learning system that utilisez neural guided program synthesis to learns to solve problems. Wüst et al. (2024a) furhter demonstrates the advantages of using program synthesis for unsupervised learning of complex, relational concepts from images, focusing on the benefits in terms of generalization, interpretability, and revisability. Stammer et al. (2024b), on the other hand investigated the benefits of symbolic representations for improved generalization and interpretability of low-level visual concepts. The position of the importance of symbols for AI explanations is further discussed by Kambhampati et al. (2022). The approach of Dinu et al. (2024) combines generative models and solvers by use of large language models as semantic parsers. Shindo et al. (2025) model human ability to combine symbolic reasoning with intuitive reactions by a neuro-symbolic reinforcement learning framework.

## 5. Alternative Views

Historically, Fodor & Pylyshyn (Fodor & Pylyshyn, 1988) argued for the emergence or implementation of symbolically reasoning structures within neural networks as a necessary aspect for achieving human-like meta-learning. However, the considerations for meta-learning discussed in their and our paper strongly focus on the learning of logical and arithmetic rules where concepts can be reduced onto symbolic expressions. These representations, therefore, naturally align well with the abilities of symbolic reasoners, but leave out other possible forms of meta-learning systems. Considering different modalities, for example for composing visual patterns or motion sequences, might pose a strong hurdle for classical symbolic systems. Such domains that do not operate over discrete 'crystallized' symbols, but rather operate on abstract 'fluid' concepts, still lack a well defined notion of what constitutes meta-learning within them. As a consequence it is unclear how to measure and systematically asses the abilities of models with regard to meta-learning in possible benchmarks.

**Untargeted Emergence of Systematic Reasoning.** Even without targeted training towards meta-learning models, LLM have shown to exhibit emergent abilities for various tasks (Brown et al., 2020; Wei et al., 2022a; Schaeffer et al., 2024). While 'true' understanding of the world might only be achieved via (embodied) interaction (Lipson & Pollack, 2000; Gupta et al., 2021; Zečević et al., 2023), some works have argued that such abilities might even be learned through mere passive observation (Lampinen et al., 2024), while other approaches argue for the value of self-explanation guided learning (Stammer et al., 2024a). Considering the underlying aspect of systematic learning and reasoning, several works already where able to distill symbolically acting *circuits* that emerged during training from LLM (Olah et al., 2020; Wang et al., 2022; Conmy et al., 2023; Hanna et al., 2024). In light of these results, it stands yet to to be seen whether meta-learning abilities of language reasoning models might also emerge as a consequence of pure scaling laws (Sutton, 2019; Kaplan et al., 2020; Bubeck et al., 2023).

## 6. Discussion and Conclusion

For this final section, we will revisit the key points that constitute our position (c.f. Sec. 1) and that we believe to form important aspects towards the goal of achieving meta-learning models capable of performing human-like systematic compositionality:

**(I) Criteria for Systematic Compositionality.** The main criteria for models with productive, systematic and compositional skills remain **compositional representation** and **structure-sentitive operations**.

**(II) Non-systematic Behavior.** As Fodor and Pylyshyn's trained model exhibits various non-systematic behaviors, it failed to demonstrate human-like compositional learning capacities and, furthermore, refutes the presented claims that their meta-learning framework is achieving human-like systematic generalization.

**(III) Assessing Compositionality.** Systematic testing of several types of out-of-distribution episodes is necessary to assess compositional skills.

**(IV) Emergence and Learning of Symbolic Representations.** Meta-learning systems have to encourage the emergence of symbolic representations during training. For that, we expect training tasks and model architecture that makes iteration, self-validation, and self-correction over the extracted rule sets possible as well as necessary.

Before concluding we now summarize the key considerations required for achieving truly meta-learning systems.

**Aspects of meta-learning.** The limitations of current neural models emphasize the importance of hybrid architectures that integrate the strengths of symbolic and connectionist paradigms. Neuro-symbolic models offer a compelling solution, combining explicit rule representation, error correction mechanisms, and dynamic scalability. Key advancements in this direction include, **(1) Systematicity**: Embedding mechanisms for representing and manipulating compositional rules within neural architectures; **(2) Reflective Reasoning**: Incorporating iterative self-correction processes to emulate human-like adaptability; **(3) Scalability**: Enabling models to dynamically expand rule sets and adapt to novel tasks, mirroring human flexibility; and we will discuss those in the following:

**Systematicity.** In this paper we argue that a key property of meta-learning systems is the ability to refrain from non-systematic errors.

**Reflection and iterative refinement.** A distinct ability of human reasoning is the ability to *reflect* and develop a set of currently hypothesized rules. The extraction and validation of rules from provided support examples might pose a complex task which can scale with exponential complexity with number of provided support examples. Models exhibiting one-shot behavior might be able to perform such tasks up to a particular problem size, but are ultimately limited by their own model capacity. In this paper we, therefore, provide arguments towards the use of reflective learners, as a particular class of models, capable of iterative refinement and self-correction of their current beliefs, a practice already adopted with great success for general language reasoning models (Wei et al., 2022b; Stammer et al., 2024a; Yao et al., 2024; DeepSeek-AI et al., 2025). This iterative behavior allows for repeated validation of the conjectures rules and, therefore, fundamentally stands in contrast to models trained to provide answers in a one-shot fashion.

**Scalability, memory and context.** An often overlooked part of learning to reflecting upon ones beliefs is the requirement of learning to store and operate on suitable representations of the models' beliefs. Particularly, this includes the presence of some sort of memory that can be read and updated. Upon the application of rules to a given query a model might additionally want to track its current context (e.g. the nesting depth of current rules), which, again, might require some sort of memory to generalize to arbitrary problem sizes and overcome the limitations of a static number model parameters.

**Conclusion.** Models with the discussed properties have the potential to address foundational critiques of connectionism while advancing the capabilities of artificial cognition. By bridging the gap between symbolic and connectionist principles, hybrid architectures could achieve systematicity and productivity, paving the way for truly human-like reasoning.

The enduring relevance of Fodor and Pylyshyn's critique underscores the challenges in developing systems capable of systematic generalization and compositional reasoning. While meta-learning frameworks represent significant progress, they fall short of resolving foundational limitations. Future advancements must embrace integrative approaches that merge the strengths of symbolic and connectionist paradigms, paving the way for a more robust understanding of artificial cognition. By addressing these challenges, we can move closer to realizing the vision of human-like artificial intelligence.

# 7. Impact Statement.

Strong meta-learning abilities show as an important skill to navigate the complex and changing tasks of today's world. When presenting models that aim to robustly adapt to novel environment, it is important to refrain from making unsolidified claims about the achievement of meta-learning systems which ultimately do not hold true upon closer inspection. Hiding behind details of what technically constitutes as meta-learning systems does not help the general discussion, but might enhance the public trust in such models, possibly leading into a false reliance in them. Our analysis showed that modern neural meta-learning systems can only archive such tasks, if at all, only under a very narrow and restricted definition of a meta-learning setup. In this paper we promote the systematic evaluation of meta-learning systems beyond their training distribution in order to truthfully assess their ability of performing compositional reasoning. We furthermore. As a result, we claim that 'Fodor and Pylyshyn's Legacy' persists and we conclude that there is *still no human-like systematic compositionality learned in neural networks* as of today.

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

# A. APPENDIX: Position: Fodor and Pylyshyn's Legacy — Still No Human-like Systematic Compositionality in Neural Networks

This appendix contains the full set of grammar rules, support examples and query examples of Lake and Baroni's meta-learning. We reevaluated their pre-trained $'net - BIML - top'$ model on the same set of $'algebraic'$ testing-episodes. Here, we reported the outputs for #133, #132, #122, and modified #1.

## A.1. Complete responses for Lake and Baroni's meta-learning testing-episodes #133.

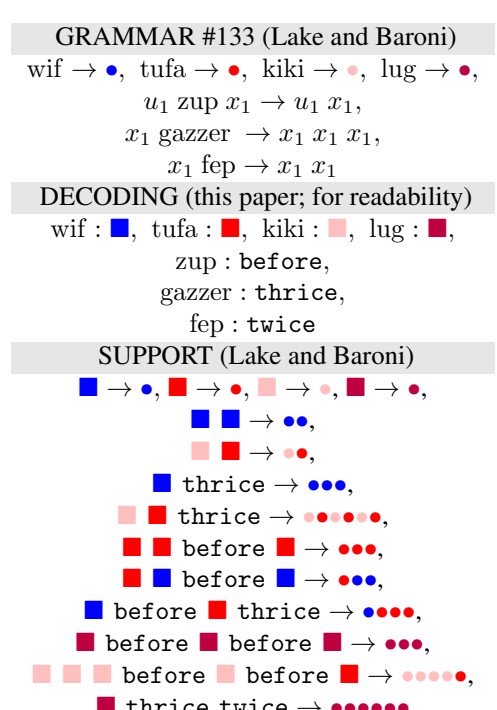

*Table 3.* GRAMMAR and SUPPORT for Episode #133; decoded for better readability. (Best viewed in color.)

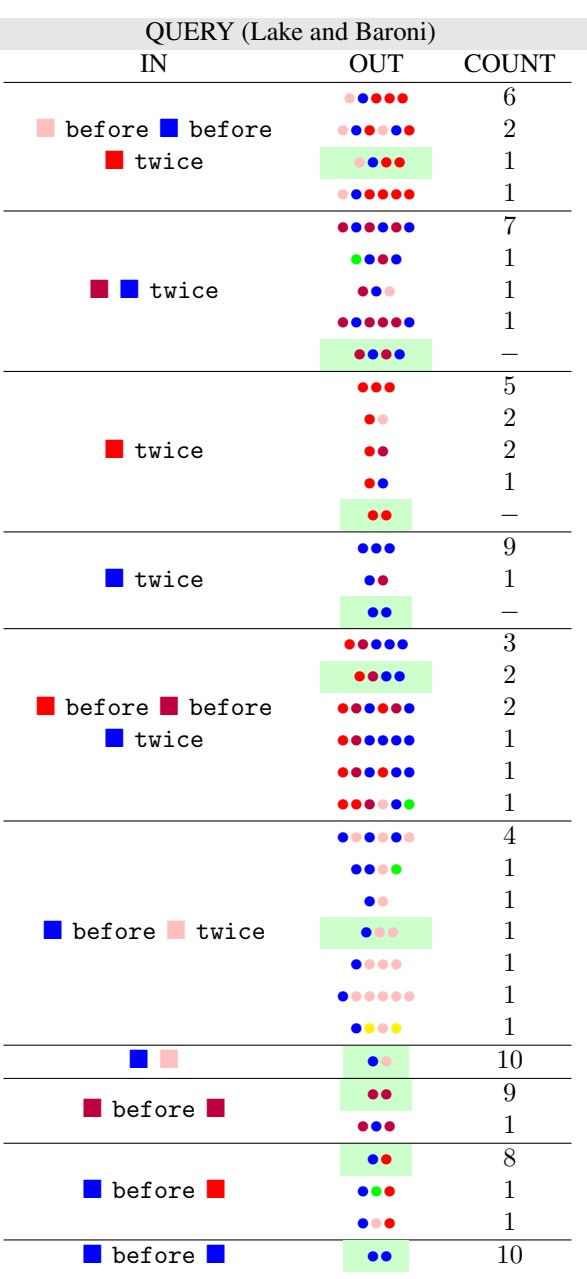

*Table 4.* Episode #133 with 10 evaluations for each query example; decoded for better readability. Expected outputs backed with green. The model shows *incoherent processing* and *systematically mistakes* twice *for* thrice. (Best viewed in color.)

## A.2. Complete responses for Lake and Baroni's meta-learning testing-episode #32.

| GRAMMAR #32 (Lake and Baroni) |
| :---: |
| tufa → •, zup → •, kiki → •, lug → •, |
| $x_1$ dax $u_1 → u_1\ x_1$, |
| $x_1$ gazzer $x_2 → x_1\ x_2$, |
| $x_1$ wif $x_2 → x_1\ x_1\ x_2\ x_2\ x_2\ x_1$ |

| DECODING (this paper; for readability) |
| :---: |
| tufa : ■, zup : ■, kiki : ■, lug : ■, |
| dax : `after`, |
| gazzer : `before`, |
| wif : `twice before and once after three times` |

| SUPPORT (Lake and Baroni) |
| :---: |
| ■ → •, ■ → •, ■ → •, |
| ■ ■ → ••, |
| ■ ■ → ••, |
| ■ ■ → ••, |
| ■ ■ → ••, |
| ■ ■ ■ → •••, |
| ■ `after` ■ → ••, |
| ■ `before` ■ → ••, |
| ■ ■ `after` ■ → •••, |
| ■ `after` ■ `after` ■ → •••, |
| ■ `after` ■ `after` ■ → •••, |
| ■ ■ `after` ■ `after` ■ → •••• |

*Table 5.* GRAMMAR and SUPPORT for Episode #32; decoded for better readability. (Best viewed in color.)

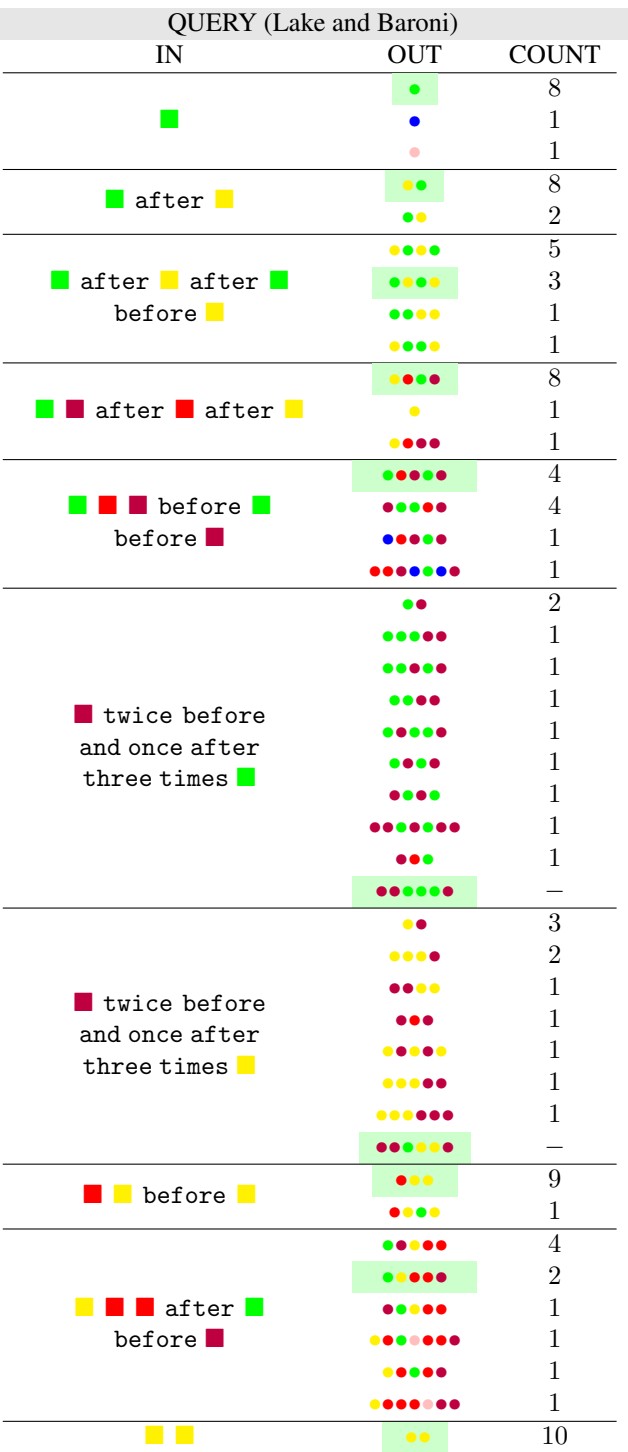

*Table 6.* Episode #32 with 10 evaluations for each query example; decoded for better readability. Expected outputs backed with green. (Best viewed in color.)

### A.3. Complete responses for Lake and Baroni's meta-learning testing-episode #122.

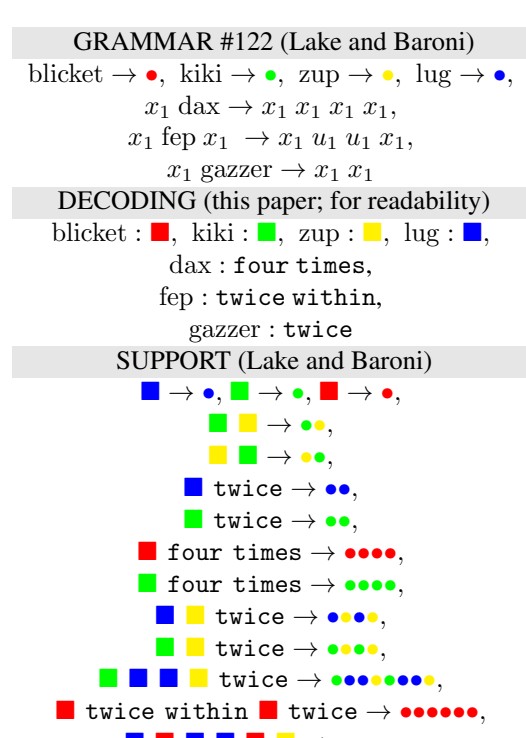

Table 7. GRAMMAR and SUPPORT for Episode #122; decoded for better readability. (Best viewed in color.)

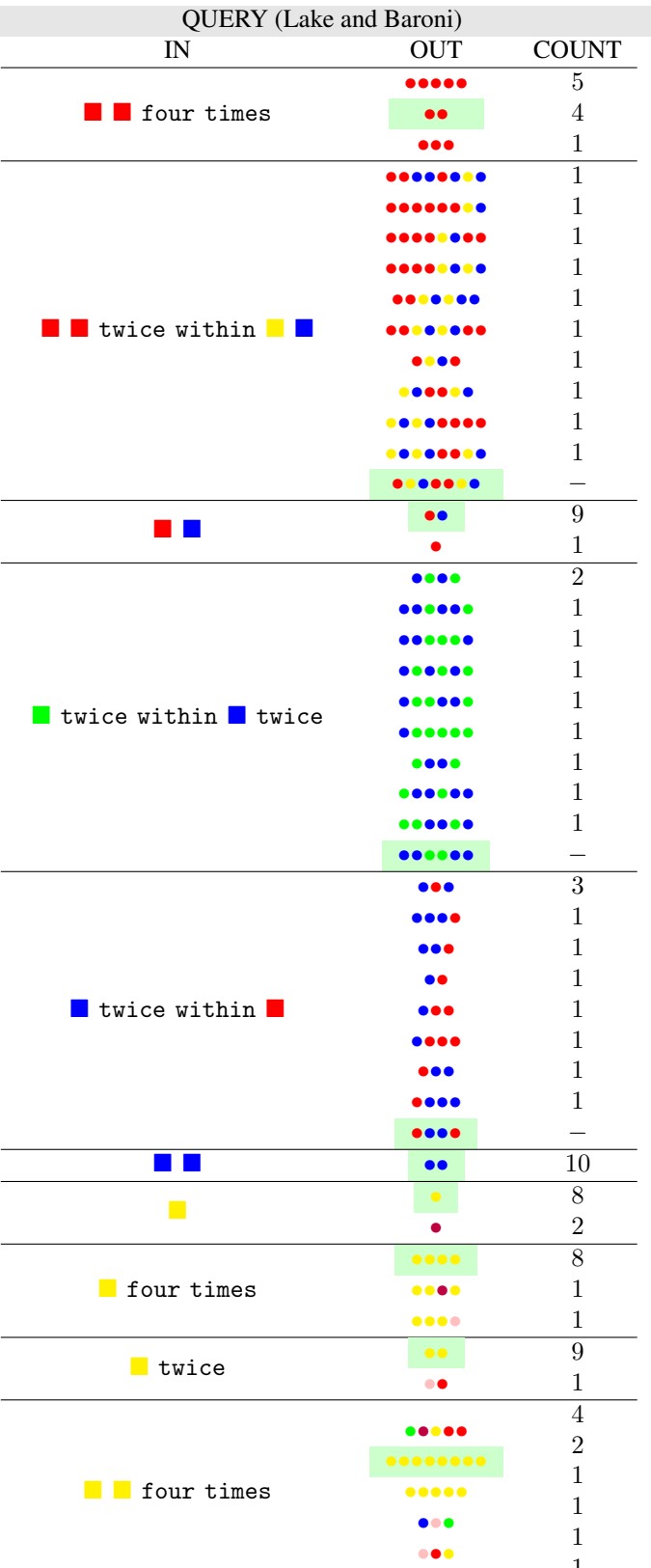

Table 8. Episode #122 with 10 evaluations for each query example; decoded for better readability. Expected outputs backed with green. (Best viewed in color.)

## A.4. Complete responses for our modified version of Lake and Baroni's testing-episode #1.

*Table 9.* GRAMMAR and SUPPORT for Episode #1; decoded for better readability. (Best viewed in color.)

*Table 10.* Our own query examples for Episode #1 with 10 evaluations each; decoded for better readability. Expected outputs backed with green. (Best viewed in color.)

