# OpenReview forum: "Position: Fodor and Pylyshyn's Legacy — Still No Human-like Systematic Compositionality in Neural Networks"
_ICML.cc/2025/Position_Paper_Track — Submitted to ICML 2025 Position Paper Track_

### Official Review · Reviewer_cY3L · 2025-03-12

**Significance:** 3
**Argument Clarity:** 3
**Rating:** 4
**Confidence:** 4

**Questions:**

Please see "strengths and weaknesses" above for discussion points I'd like to raise.

I am pretty sure line 385 should say "Lake and Baroni's" not "Fodor and Pylyshyn's".

**Discussion Potential:**

4

**Paper Summary:**

This paper presents detailed evidence that Lake and Baroni 2023 (Nature; LB23) have over-stated the degree to which their small encoder-decoder networks trained with metalearning have learned to generalize systematically. The paper couches these criticisms in a broader discussion of the extent to which we currently have evidence that neural networks have learn to generalize systematically.

## update after rebuttal

I remain supportive of accepting this work. I would urge the authors to consider also submitting a note to Nature.

**Position:**

Yes

**Position In Title:**

Yes

**Related Work:**

3

**Strengths And Weaknesses:**

1. This seems like an important paper. Assuming the analyses in the paper are correct, LB23 contains some oversights and errors that call their findings into question. This alone warrants publication, though one might think that Nature is a better venue for such a corrective than ICML.

2. The only aspect of the paper that gives me pause right now is the leap from the LB23 criticisms to broader conclusions about what neural networks have or have not achieved when it comes to systematicity. The models from LB23 are small, highly idealized artifacts that are trained in highly unrealistic settings that may not have much to do with systematicity in the wider world. Thus, we really jump from from this setting to the claim "Still No Human-like Systematic Compositionality in Neural Networks" or to the negation of that claim.

3. To expand on the above: scenarios in the LB23 paper are not realistic, and the blank slate learners involved are not relevant. Humans can solve these tasks because of their vast experience with pattern matching and problem solving. My conviction about this also makes me want to push back against the sweeping conclusions made in this paper on the basis of what looks like a negative finding in LB23.

4. In my view, the largest and best LLMs that we interact with today show substantial evidence of systematicity. They can do all sorts of novel tasks and handle lots of very rare constructions. I suspect that this is because they have developed highly structured representations. I can't show this because I have only behavioral access to most of these models, but smaller open-source ones really do support this contention. Here are a few papers in support:

    * Language Models Learn Rare Phenomena from Less Rare Phenomena: The Case of the Missing AANNs
    https://aclanthology.org/2024.emnlp-main.53/

    * CausalGym: Benchmarking causal interpretability methods on linguistic tasks
    https://aclanthology.org/2024.acl-long.785/

    * Large linguistic models: Investigating LLMs’ metalinguistic abilities
    https://arxiv.org/abs/2305.00948

   Any human that showed the level of behavioral systematicity seen in these papers would be regraded as fluent in the relevant languages. Many humans do not rise to this level behaviorally, in fact, at least in experimental settings.

5. It's striking to me how old-fashioned Fodor and Pylyshyn's views sound now. The idea that neural network representations are a priori "irrelevant to psychological theory" seems to be question begging -- one can only say this if one is taking as axiomatic that mental representations are symbolic in the particular sense that they have in mind.

**Support:**

3

---

> ### Author Rebuttal · Authors · 2025-03-28
>
> Dear Reviewer cY3L,
> Thank you for your thoughtful review and for highlighting the importance of our position paper and its discussion potential.
>
> **W1: \[Leap from LB23 to broader conclusions\]** In our position paper, we not only critique LB23's claims, but also outline a more general approach to arguing, testing, and training for systematic generalization or compositionality, taking into account F&P's arguments against systematicity in plain neural networks. To discourage overstating legitimate and important but gradual progress towards systematic compositionality, we demonstrate the relevance of our outline with an effective critique of L&B's Nature paper.
>
> **\[Comment on challenging our position\]** We agree with the reviewer that our position has much potential for discussion, especially with respect to less realistic or more formal meta-learning tasks such as in LB23. We also believe that this position poses a clear challenge to future claims of systematic generalization, and can lead to fruitful and constructive discussion of more substantial claims in future publications.
>
> **\[Comment on Evidence of systematicity\]** Thank you for pointing out other interesting papers that oppose our position. We have included them in the discussion of alternative views in the final version.
>
> **\[Comment on Fodor and Pylyshyn's view\]** We are also fascinated by F&P's old-fashioned view of the relevance of neural network representations. However, we also find it interesting how their arguments fit with the observation that simple neural networks without any kind of built-in symbolics often struggle with systematic composition tasks. Indeed, the training of more complex models has become much more sophisticated and may overcome the limitations that F&P saw in purely statistical methods. For example, it is debatable whether the self-attention mechanism, as well as the more recent test-time scaling approaches, blur the boundaries between connectionism and symbolism drawn by F&P.
>
> Please let us know if there is any additional information we can provide to further clarify our contributions. Thank you once again for your time and effort in reviewing our paper!
>
> Sincerely,
> Authors

---

### Official Review · Reviewer_v6xv · 2025-03-13

**Significance:** 2
**Argument Clarity:** 1
**Rating:** 2
**Confidence:** 3

**Questions:**

Do you have examples on non systematic errors of concern to illustrate claim 1?

(Also in weakness)
What does it mean that some test examples involve a combination of non primitive operations already in training (line 190)? Why did L&B not catch these training exposures, or what is different about the combinations being analyzed here that considers more examples to be from training?

What does it mean to extract, validate, and correct? (291) What would this look like and why is it necessary for systematic reasoning? Multiple references to these requirements which are not defined or justified.

What is the basis for dismissing systematicity as a concern in these systems? Why should we focus on productivity?

**Discussion Potential:**

2

**Paper Summary:**

Through experiments and examples, this paper argues that the meta learning proposed by lake and baroni (L&B) fails to provide consistent human like systematic compositionality. Furthermore, they argue that it lacks the verification properties needed for such behavior and that systematicity is a less important focus than productivity.

## update after rebuttal

Some of my concerns have been addressed by the authors, so I have raised my score. I have not raised it to an accept because my primary issue (this should be a primary track paper with full details about its experiments, not a position) still stands.

**Position:**

No

**Position In Title:**

Yes

**Related Work:**

3

**Strengths And Weaknesses:**

The major problem with this paper in the position track is that it is not a position paper, except in the sense that all good papers are position papers because they argue for a thesis on the basis of their experiments. While the point they have to make is potentially valuable to the relevant cog sci community, neither the experimental results nor the argumentation provide enough detail to make their position convincing. I believe this problem is due to the unnatural framing of their findings as a position paper, rather than an experimentally rigorous argument with alternatives fully tested.

The natural, best form of this paper would involve an alternative proposal oriented around their stated requirements for compositional reasoning. It would require them to test their proposal. Another option is to focus only on the empirical criticism of L&B by simply expanding on their discussion of its specific failures.

Lack of details about experiments. What does it mean that they involve a combination of non primitive operations already in training (line 190)? Why did L&B not catch these training exposures, or what is different about the combinations being analyzed here that considers more examples to be from training?

What does it mean to extract, validate, and correct? (291) What would this look like and why is it necessary for systematic reasoning? Multiple references to these requirements which are not defined or justified.

It’s unclear why systematics is irrelevant. Classic examples like the “horse riding an astronaut” in generative image models make it clear that even recent models aren’t necessarily capable of systematic reasoning, so why would the authors prefer productivity tests at the expense of systematicity?


Position on evaluation: requirement for internal representation analysis should be argued for on the basis of specific problems and its evaluations. For the OOD tests, is that not what L&B are directly attempting?

MINOR:
“That were” 281, combination of emdash and comma parenthetical in 281, “challange” in 055.

**Support:**

1

---

> ### Author Rebuttal · Authors · 2025-03-28
>
> Dear Reviewer v6xv,
> Thank you for your thoughtful review. We will address your concerns in the following:
>
> **W1: \[Not a position paper\]** While we argue that neural networks have not yet achieved human-like systematic compositional generalization, we also outline how to argue, test, and train for systematic generalization or compositionality given the limitations raised by F&P. To discourage overstating the legitimate and important but gradual progress toward systematic compositionality, we demonstrate the relevance of our outline with an effective critique of L&B's Nature paper. L&B's limitations are thus discussed only as evidence for our position on how to approach the assessment of human-like systematic compositionality, and we do not frame some findings about L&B as a position, but rather, we have stated a position and exemplified it through these investigations. We therefore take a clear position on *how to improve the way we conduct and evaluate machine learning research* on human-like systematic compositionality, which is in line with the ICML call for position papers.
>
> **W2: \[Alternative proposals\]** We disagree that our work is not valuable as a position paper critiquing current approaches and claims to systematic compositionality. While a focus on empirical evaluation or criticism could be considered, we see our approach as a more constructive contribution to fruitful discussions and future research on systematic compositionality.
>
> **Q1: \[Examples of non-systematic errors\]** The paragraphs *Non-systematic parsing* (224) and *Violating structure-sensitivity* (245) discuss in detail examples of errors in L&B's MLC setup that illustrate claim 1. In failing to extract the correct rules, the system is inconsistent in its behavior, applying different parsing schemes for examples of the same validation episode. And for some examples where plausible systematic errors would overlap in similar outputs, the system even shows different behavior.
>
> **W3+Q2: \[Combination of non-primitive operations\]** L&B’s meta-learning episodes (for training and validation) consist of a combination of four primitive operations $u \rightarrow c$, mapping a given pseudo-language token $u$ to a color token $c$, and three non-primitive (unary or binary) operations $v_1u \rightarrow f_u(v_1)$ or $v_1 u v_2 → g_u(v_1,v_2)$ that repeat, permute and/or concatenate surrounding tokens (or token strings). Similar to output colors $c$ having different “names” $u$ in different episodes, the same non-primitive operation $f_u$ (or $g_u$) can have different “names” $u$ (i.e. $f_u=f_{u’}$ for $u \neq u’$). As L&B note themselves, every single non-primitive operation in the validation was already seen during training but in possibly different combinations with other operations. In addition to that, we found that if we do not account for different names of operations, 179/200 validation episodes have a combination of non-primitive operations with at least one similar operation in the training set. In these cases, the model may have learned a shortcut pattern and simply extracted the new name for the familiar operation. We will inlcude the script for this analysis in the appendix. The fact that L&B did not properly address this issue makes it important to address the need for proper OOD testing when claiming to assess compositional skills.
>
> **W4+Q3: \[Extract, validate, and correct extracted rules\]** At this point we are talking about the representation (of the current belief) of the extracted rules. L&B do not attempt any internal representation analysis because they approach the transduction task with a black-box transformer model and only compare input-output results with human behavior. We argue that non-reflective training, in the sense of training that does not allow or encourage iteration on the extracted rules, will fail to develop systematic extraction abilities.
>
> **W5+Q4: \[Dismissing systematicity\]** There seems to be a misunderstanding here. At no point do we advocate abandoning systematicity in favor of productivity. On the contrary, we argue for a "focus on systematicity rather than productivity" (297), because mere productivity testing only reaches the contextual limits of models without necessarily testing their compositional capabilities within those limits. Thus, we fully agree with the reviewer that systematicity is important.
>
> Please let us know if there is any additional information we can provide to further clarify our contributions. Thank you once again for your time and effort in reviewing our paper!
>
> Sincerely,
> Authors

---

> > ### Comment · Reviewer_v6xv · 2025-04-04
> >
> > My concern is that the primary contribution of your paper is an empirical result. I am not convinced it belongs in the position paper track, because it is not a broad critique; it is an alternative experimental method reaching a different empirical result. Yes, you take a position as to the appropriate experimental practice. Yes, you take a position as to the meaning of your findings. But these positions should be represented in every empirical paper submitted through any track to any conference. Why did you send an experimental paper, then, to the position track?
> >
> > Thank you for clarifying the meaning of "combination of non primitive operations already in training." I hope you will make it a little clearer through explicit definitions and examples. I agree with you that it is likely that the model is mapping compositions from training onto new surface forms, and would not generalize without exposure to alternative forms of those compositions. However, you have not shown further evidence that this is, in fact, happening. For example, are the equivalent compositions representationally similar when the model succeeds at handling them? And if there are 21 genuinely novel combinations, at what rate does the model fail at those?
> >
> > I still don't understand what, specifically, it means to extract, validate, and correct.  If this is a key part of the position that justifies this as a position paper, I expect it should have some detail as to your specific recommendations and specific examples of each.
> >
> > Thank you for clarifying that we should focus on systematicity. I'm sorry I misread the meaning there.
> >
> > In summary, however, I still think this is an empirical analysis / experiment paper, not a position paper. The experiments are not quite expanded enough, especially lacking a validation of the idea that the model is learning superficial mappings onto known primitive combinations. Wouldn't it require systematicity even to correctly map the primitive surface forms onto the appropriate combinations? What would a superfically systematic shortcut look like, in contrast with true systematicity? How can you test for it?
> >
> > Overall, I feel that this paper could be worked into a main track paper by expanding the experiments, but I don't understand why it's a position paper, unless the original L&B paper also was a position paper.
> >
> > I'll move my score up to reflect your clarifications and hopefully some future edits to clarify the paper. I still don't think it should be accepted.

---

> > > ### Author Response · Authors · 2025-04-08
> > >
> > > We thank the reviewer for their continued engagement and for raising their score. However, we disagree with the view that the primary contribution of your paper is an empirical result, since we do not bring any a posteriori argument or method how to solve systematic compositionality with artificial cognition. Recapitulating the historical arguments of F&P, we have worked out that *compositional representations* and *structure-sensitive operations* are the key properties of systems with compositional abilities. And while we present an empirical critique of L&B's work, which reveals a violation of structure-sensitivity with their model, we use this finding to outline necessary aspects of training and evaluation of meta-learning systems to achieve and assess their systematicity, taking into account the relevance of compositional representations and structure-sensitive operations. Our empirical contribution is thus only to the method of refuting L&B's claims, but our position is not limited to their approach. Since the other reviewers also see the relevance and discussion potential of our contribution, we ask the reviewer to recapitulate what prevents them from acknowledging our position.
> > >
> > > Regarding the other questions:
> > > * **[combination of non-primitive operations already in training]** We have added a reference to a new section in the Appendix that clarifies and discusses the relevance of combinations of non-primitive operations. This potential problem is somewhat orthogonal and implicitly addressed by our findings of non-systematic behavior and violations of structure sensitivity, as it concerns the informative value of good performance with respect to the systematicity involved. We directly show non-sensitivity even within episodes that do not contain new combinations of non-primitive operations (only episode #122 in the appendix has a new combination). We raise this point only to highlight some a priori lost potential of L&B's work.
> > > * **[extract, validate, and correct]** This is not a key part of our position, as it is only one type of mechanism we propose to achieve a system with compositional representations, and we consider completely different approaches possible, but we agree that an example is helpful, and have added some explanation and an example of neuro-symbolic explanatory interactive learning [1] to the paragraph.
> > >
> > > Sincerely,
> > >
> > > Authors
> > >
> > > [1] Stammer, Wolfgang, Patrick Schramowski, and Kristian Kersting. "Right for the right concept: Revising neuro-symbolic concepts by interacting with their explanations." Proceedings of the IEEE/CVF conference on computer vision and pattern recognition. 2021.

---

### Official Review · Reviewer_53mC · 2025-03-14

**Significance:** 3
**Argument Clarity:** 2
**Rating:** 4
**Confidence:** 3

**Questions:**

Questions, comments, and suggestions/typos.

Please clarify on the difference of your work from earlier work of Goodale & Mascarenhas, 2023 ,
that you cite and that also points to the limits of the work of Lake and Baroni.

(makes a stronger what?) line 117: ".. of A or B. In fact, systematicity makes a stronger by using
118 a weaker assumption as, ”[p]roductivity arguments infer.."

typo, 366:.. that utilisez neural guided program synthesis to learns to solve
problems. Wu ̈st et al. (2024a) furhter demonstrates the


130, 'EPISODES' (on 1st reading I thought it was a system, but I now
think it is just 'several episodes'): ... work with EPISODES associated
with different transduction grammars. Each episode combines a SUPPORT
set of input-output..


154 ('but directly address ..'), section 3, awkard/usage: ".. and
Pylyshyn’s arguments Lake and Baroni are referring to, since they
primarily present an implementation of what they claim is a human-like
systematic capability, but directly address a challenge. They
themselves situate their work .. "


134 reword: 'while any pair is consistent with the associated
grammar.' (eg 'It is the case that any pair is ..', or 'with all pairs
being consistent with the associated grammar')

210, word it or structure it differently: "and will therefore refer to
as ⟨twice⟩ and mixes it up with the token ⟨gazzer⟩"

300 ('Position on Evaluation'): 'test' instead of 'testing' in
"network architecture, or the use of comprehensive ablation studies
that systematically testing a model’s behavior in out-of-distribution
situations."

322: 'possibility iterate, validate and correct over the extracted rule
sets.'


(here, I believe the authors mean Lake and Baroni here, not
Fodor&Pylyshyn.. )

385 (II) Non-systematic Behavior. As Fodor and Pylyshyn’s
386 trained model exhibits various non-systematic behaviors, it
387 failed to demonstrate human-like compositional learning
388 capacities and, furthermore, refutes the presented claims
389 that their meta-learning framework is achieving human-like
390 systematic generalization.

**Discussion Potential:**

3

**Paper Summary:**

The authors explain the systematicity critique of NN approaches made by Fodor and Pylyshyn (Human-like Systematic Compositionality), and attempts by researchers (Lake and Baroni) to address certain aspects via meta learning techniques, and explain why existing approaches still fall well short of the goal (the generalization capabilities of humans).


## update after rebuttal

I have read all the reviews and authors' rebuttals.  I think the authors can improve the paper and reduce the clarity issues. The topic
is a great one for a position paper (to bring up a discussion of the subtleties, what F&P meant [was it clear enough?!], and so on). I'll keep my rating as 'Accept'.  Thanks.

**Position:**

Yes

**Position In Title:**

Yes

**Related Work:**

3

**Strengths And Weaknesses:**

Strengths.  Illuminates the subtle challenges that remain.

Weakness: a number of typos and clarity issues, though overall understandable.

**Support:**

3

---

> ### Author Rebuttal · Authors · 2025-03-28
>
> Dear Reviewer 53mC,
> Thank you for your thoughtful review and suggestions. We have corrected the typos you spotted in the final version.
>
> **Q1: \[Goodale & Mascarenhas, 2023\]** In contrast to Goodale & Mascarenhas, 2023, which is a direct response to Lake and Baroni, 2023, our position paper has a much broader scope. It is more systematic in assessing the limitations of MLC, as we not only argue that neural networks have not yet achieved human-like systematic compositional generalization but also outline in section 3.2 how to argue, test, and train for systematic generalization or compositionality, taking into account F&P’s arguments against systematicity in plain neural networks. This can be seen for instance in paragraph (304) *Position on Implementation and Training.* Our effective critique of L&B's Nature paper thus further illustrates the relevance of our proposed outline.
>
> **Q2: \['EPISODES'\]** Yes, it is just 'several episodes associated with different transduction grammars'. We changed the emphasis to lowercase italics for less confusion.
>
> Please let us know if there is any additional information we can provide to further clarify our contributions. Thank you once again for your time and effort in reviewing our paper!
>
> Sincerely,
> Authors

---

### Official Review · Reviewer_PjTR · 2025-03-14

**Significance:** 2
**Argument Clarity:** 3
**Rating:** 3
**Confidence:** 4

**Questions:**

Figure 1 is quite confusing. It is unclear what information the authors intend to convey with this figure, and it does not seem to be connected to the rest of the paper.

**Discussion Potential:**

2

**Paper Summary:**

This paper argues that neural networks still fail to achieve human-like systematic compositional generalization. To support this view, the authors assess Lake and Baroni’s claim and highlight that their MLC framework lacks key elements necessary for making substantive assertions about systematic generalization. The experimental results reinforce the authors' argument.

**Position:**

Yes

**Position In Title:**

Yes

**Related Work:**

3

**Strengths And Weaknesses:**

### Strengths:

1) This paper is well-written overall. The authors clearly outline the history of studies on compositional generalization and recent works, providing a strong background.
2) The authors present substantial evidence demonstrating the limitations of the MLC framework.

### Weaknesses:

1) The paper primarily focuses on critiquing Lake’s work, which limits its overall contribution.
2) While the MLC framework may not fully address all systematic generalization (SG) problems, it has been shown to exhibit human-like behavior in certain tasks. Identifying cases where MLC fails does not seem particularly valuable, as there are many other SG challenges it cannot solve, such as compositional generalization in text-to-image generation.
3) Most researchers do not yet believe that SG has been fully resolved, as evidenced by the continued emergence of new papers addressing SG issues. The authors' argument is restricted to a narrow scope.

**Support:**

3

---

> ### Author Rebuttal · Authors · 2025-03-28
>
> Dear Reviewer PjTR,
> Thank you for your thoughtful review and for highlighting our strengths, including the strong background of our position outlining the history of studies on compositional generalization and recent works. We will address your concerns in the following:
>
> **W1+3: \[Limited scope\]** We politely disagree with the argument about the limited scope of L&B's work. To clarify, while we argue that neural networks have not yet achieved human-like systematic compositional generalization, we also outline, e.g. in Section 3.2, how to argue, test, and train for systematic generalization and compositionality from a more general point of view, taking into account F&P's arguments against systematicity in plain neural networks. Moreover, while we agree that parts of the research community are investigating and identifying further SG issues, we also observe that a large fraction of recent papers claim to have solved a crucial part of SG (e.g., \[1\],\[2\],\[3\],\[4\],\[5\]). Therefore, to discourage overstating legitimate and important but gradual progress towards systematic compositionality, we demonstrate the relevance of our outlined position with a case of effective critique of L&B's Nature paper. We agree that this structure could be made a bit clearer to the reader in our paper. We have now added a clarifying paragraph in the introduction before listing our contributions. Thank you for pointing this out!
>
> **W2: \[MLC exhibiting human-like behavior\]** We acknowledge that MLC shows human-like behavior in certain tasks, but we argue that these tasks were insufficient to claim MLC's productivity or systematicity in its target domain. One of our arguments is that L&B's results can be explained by mere statistical (meta-)pattern matching. Our counter-examples also contradict the claim of human-like systematic generalization because they are less demanding for humans and lead to expect different human behavior compared to MLC. The value of using this case as an illustration is to point out the limitations of a paper that claims to solve aspects of SC, when in fact our analysis shows that this is not the case. This proves the point that it is immature to claim to have (partially) solved SC. Thus, we do not criticize L&B's paper for not solving all aspects of SC.
>
> **Q: \[Figure 1\]** The goal of the figure is to illustrate the mismatch between state-of-the-art AI models and human capabilities in systematic instruction following. We agree that the caption is suboptimal; we have changed it to: "Even when prompted to generate an image without text, recent AI models fail to do so, indicating a limited systematic generalization to the semantics of the word *text*."
>
> Please let us know if there is any additional information we can provide to further clarify our contributions. Thank you once again for your time and effort in reviewing our paper!
>
> Sincerely,
> Authors
>
> \[1\]  Brown, T. et al. *Language Models are Few-Shot Learners.* Advances in Neural Information Processing Systems, 2020.
>
> \[2\]  Kaplan, J. et al. *Scaling laws for neural language models.* arXiv:2001.08361, 2020.
>
> \[3\]  Wei, J. et al. *Emergent Abilities of Large Language Models.* Transactions on Machine Learning Research, 2022.
>
> \[4\]  Bubeck, S. et al. *Sparks of artificial general intelligence: Early experiments with gpt-4.* arXiv:2303.12712, 2023.
>
> \[5\]  Srivastava, A. et al. *Beyond the Imitation Game: Quantifying and extrapolating the capabilities of language models.* Transactions on Machine Learning Research, 2023.

---

> > ### Comment · Reviewer_PjTR · 2025-04-05
> >
> > The response to W1,W2,W3 seems reasonable to me. However, I’m still concerned about whether the proposed revisions will fully address the issue. Clearly defining the scope of the paper and highlighting its key contributions to the community is a crucial aspect of any strong submission.
> >
> > Regarding Q1, I remain unsure about the relevance of including this figure, especially since the paper places little emphasis on text-to-image generation.
> >
> > If the authors can **provide a more detailed and actionable revision plan** that effectively addresses these concerns, I would be willing to raise my score.

---

> > > ### Author Response · Authors · 2025-04-08
> > >
> > > We thank the reviewer for their continued engagement. We gladly provide a more detailed revision plan below, where we provide updated versions of different sections below:
> > >
> > > ### Introduction (end of):
> > > Thus, we argue in this paper that: *Neural networks have not yet achieved learning systematic compositional skills*. Based on a case of effective critique of Lake and Baroni's framework, we outline how to argue, test, and train for systematic generalization and compositionality and demonstrate the relevance of our position.
> > >
> > > We develop this position as follows:
> > >
> > > * We identify Fodor and Pylyshyn's main arguments in the context of the compositionality challenge for artificial neural networks and locate the nature of Lake and Baroni's approach in refuting Fodor and Pylyshyn's claims that neural networks cannot reliably develop *compositional representations* and *structure-sensitive operations*.
> > > * We show that within their setup, the model exhibits various *non-systematic* behaviors that can not be considered human-like and clearly violates structure-sensitivity.
> > > * We argue for several necessary aspects of training and evaluation of meta-learning systems to achieve and assess their systematicity, taking into account the relevance of compositional representations and structure-sensitive operations.
> > > * We adapt Fodor and Pylyshyn's arguments in light of the modern development of deep-learning systems to argue for a future of models capable of learning symbolic representations for artificial cognition and representation learning.
> > >
> > > ### Section 3 (Systematicity through Meta-Learning) new introduction paragraph:
> > > In the following, we exemplify the limits of current neural network approaches by examining the systematicity achieved by Lake and Baroni's meta-learning approach. After revealing a severe lack of compositionality in their framework, we propose how to better test and train for systematic generalization and compositionality with meta-learning systems. We hereby highlight the ongoing challenges associated with compositional representations and structure-sensitive operations.
> > >
> > > ### Section 6 (Discussion and Conclusion) updated first paragraph of Conclusion:
> > > While the importance of *compositional representations* and *structure-sensitive operations* for human-like systematicity remains, the previous consideration allows the training and testing of artificial neural networks that encourage the development of such properties.
> > > By bridging the gap between symbolic and connectionist principles, hybrid architectures may be particularly promising, since they do not suffer from the limitations of neural networks without symbolic machinery as specified by Fodor and Pylyshyn.
> > >
> > > ### Figure 1
> > > We understand the concerns regarding the figure and have now created a novel figure to more clearly illustrate the position and outline of this paper. The caption of the figure reads: **The challenge of claiming and testing systematic compositionality.** Given the undisputed importance of *compositional representations* and *structure-sensitive operations* for systematic compositionality, their evaluation remains crucial and challenging. While structure-sensitivity can be assessed by comprehensive OOD testing, the investigation of representations requires some inspectable model architecture.
> > > The new figure can be found [here](https://anonymous.4open.science/r/meta-learning-figure-F66B/figure1.pdf). We agree that this helps understand the highlevel idea of our work much better.
> > >
> > > Overall, these revisions make the contributions and scope of the work much more clear and explicit. Thank you for the suggestion.
> > >
> > > Sincerely,
> > >
> > > Authors

---

### Decision · Program_Chairs · 2025-04-30

**Decision:**

Reject (with encouragement)

**Comment:**

Your paper was favorably reviewed, and we would have liked to accept it this year.  However, due to constraints on conference capacity, we had to reject some papers despite positive reviews.  The area chair's original meta-review (copied below) is a testament to the strengths of your paper.

----
This paper proposes an interesting position that is highly relevant to both ML and cognitive science. 3 out of 4 reviewers voted for acceptance, and one reviewer voted for rejection. The main criticism was that the paper was narrowly scoped, being primarily an empirical rebuttal to Lake and Baroni. This criticism does have merit: position papers should be a broad critique rather than a rebuttal to particular papers.

However, this AC believes that it is possible that this criticism can be addressed in the camera ready. The authors have already proposed these changes in their rebuttal, and the AC believes that they meaningfully address this criticism. The authors are encouraged to incorporate these changes in the camera ready.